# Evolution of the Terminal Plane from Deciduous to Mixed Dentition

**DOI:** 10.3390/children10101708

**Published:** 2023-10-20

**Authors:** María Eugenia Cabrera-Domínguez, Antonia Domínguez-Reyes, Antonio F. Galan-Gonzalez

**Affiliations:** Departamento de Estomatología, Universidad de Sevilla, 41009 Sevilla, Spain; mcabrera@us.es (M.E.C.-D.); agalan@us.es (A.F.G.-G.)

**Keywords:** terminal plane, canine occlusion, molar occlusion, deciduous dentition, early mixed dentition, permanent dentition

## Abstract

(1) Introduction: Correct development and growth of the dental arches and occlusion in the deciduous dentition is crucial for physiological occlusion in the permanent dentition. The present study evaluates the evolution of the terminal plane and canine occlusion class in the same children from deciduous to mixed dentition. (2) Materials and methods: The study included 257 children (164 girls and 93 boys) aged 3–5 years in the first phase and 8–10 years in the second phase. The chi-square test was used for the comparison of qualitative variables, while analysis of variance (ANOVA) or the Mann–Whitney U-test, Kruskal–Wallis test, and Wilcoxon test were used in the case of quantitative variables, as applicable. Statistical significance was considered for *p* < 0.05. (3) Results: The most common terminal plane in the first phase of the study was a bilateral flush plane (70%), followed by distal and mesial, with few differences between them. In the second phase, the most common terminal plane was mesial, followed by bilateral flush and distal. There were no statistically significant differences according to gender. Canine occlusion in the first phase was predominantly bilateral class I, followed by class II and class III. Similar results were recorded in the permanent dentition, except for a lesser percentage of children with canine class II. Molar occlusion in the second phase was predominantly class I, followed by half cusp class II and full cusp class II and class III. (4) Conclusions: The present study shows that knowing the age range in which maximum dental development and growth in both arches occurs may contribute to avoiding malocclusions and the possible need for orthodontic-orthopedic treatment, resulting in improved outcomes and greater stability.

## 1. Introduction

Correct development and growth of the dental arches and their occlusion in the deciduous dentition is crucial for subsequent normal physiological occlusion in the permanent dentition, since the latter is conditioned by what happens during the deciduous dentition phase [1].

From the appearance of the first teeth and their eruption, children gradually develop the dental arches and their form of occlusion. The changes that occur in these arches are influenced, apart from the genetic load, by different dietary habits—such as the consistency of food, which has changed throughout history—as well as by swallowing, breathing, phonation, and/or tongue position [2,3]. These factors mainly have an influence transversally, although they can also indirectly influence the anteroposterior growth of the arches; however, it is proximal caries in the posterior sectors or premature loss of primary molars that can play an important and direct role in the anteroposterior occlusion [4,5].

All these elements condition the generation of a mesial step, distal step, or flush terminal plane in the deciduous molars, which in turn conditions permanent occlusion (canine and molar relation, and the terminal plane of the child). Therefore, knowing how the molar and canine occlusion is in the primary dentition is important to predict the possibility of a correct molar and canine occlusion in the permanent dentition [6,7,8,9,10].

A molar class I relation in the permanent dentition is necessary for proper static occlusion, as well as benefits functional occlusion. We know that the initial molar relation depends on the terminal plane (flush terminal plane, mesial step, or distal step) of the deciduous second molars. This conditions the relation of the permanent first molars, which is established early in 25% of all children (mesial step to class I), while 50% present a cuspid-to-cuspid relation (flush terminal plane to half cusp class II), and the remaining 25% have more severe presentations (distal step to class II) [11,12].

Although few articles have analyzed the evolution of the anteroposterior relation (terminal plane and canine relation) from the deciduous dentition to the mixed dentition, a study has been carried out among preschool children in India, analyzing the terminal plane, among other parameters, and showing a flush terminal plane to be the most frequent presentation—though with a statistically significant increase in the presence of the mesial step [13].

Given the importance of a correct analysis of occlusion in the primary dentition for the diagnosis of future malocclusions and occlusal stability in the permanent dentition, and given that few studies have studied the evolution of occlusion in the anteroposterior direction from the deciduous dentition to the mixed and/or permanent dentition, the aim of this study was to longitudinally evaluate the changes that occur in the terminal plane from the primary dentition to the mixed dentition.

## 2. Material and Methods

The initial sample of this longitudinal study was obtained from 18 randomly selected schools in the city of Seville (Spain) with preschool education, and representative of the three socioeconomic levels (low, medium, and high). Currently in Seville, there are 111 kindergarten and primary schools. We divided these schools by socioeconomic status, considering private schools as high socioeconomic status, charter schools as medium status, and public schools as low status. We chose three schools at random from each of the six districts of Seville, each representative of a socioeconomic status. Once the study was conducted, we reassigned students to socioeconomic status based on parents’ level of education and occupations.

An analysis was made of the evolution of the deciduous dentition parameters (longitudinal study) with the purpose of improving our knowledge of the development and changes of the terminal plane, occlusion, and possible relations in the mixed and permanent dentition. It should be mentioned that although the initial sample consisted of 1182 children with deciduous dentition and a mean age of 4.68 ± 0.72 years (first phase of the study), in the second phase, characterized by mixed and permanent dentition, the sample was reduced to 257 children. Of these, 164 were girls (63.8%) and 93 were boys (36.2%), with a mean age of 8.3 ± 0.8 years.

Thus, of the initial total preschool children analyzed in 2017 and randomly selected from the different schools, we were able to again explore 262 children, obtaining information corresponding to both phases (1 and 2)—though 5 cases failed to meet the inclusion criteria and were excluded.

Inclusion criteria:Children that had completed the first phase of the study in the deciduous dentition.Children currently in the second phase of the study, with mixed or permanent dentition and aged 8–10 years.Children with no past or present orthodontic or orthopedic treatment.No proximal caries in posterior sectors.No premature loss of any primary molar.

Exclusion criteria:Patients failing to meet some inclusion criterion or presenting any other circumstance capable of altering the study analysis.

The study was approved by the Biomedical Research Ethics Committee of the Government of Andalusia (Ref. code: 0937-N-15).

The parents of the participating children were informed about the purpose of the investigation, and written consent was obtained from them before starting the study. After the end of the study, the parents or caregivers received a report detailing the diagnosis, offering health education information, and providing recommendations for solving the possible problems detected.

The materials commonly employed in dental explorations were used: intraoral mirrors, dental probes, tongue depressors, antiseptic solution, material containers, gloves, masks, paper towels, and calibrators (maximum precision (±0.1 mm) calipers). In order not to complicate or interfere with the school timetable, in both study phases, the moment of the exploration (day and time) was pre-established, and the pupils were examined in small groups in a room destined for the purpose and under monitoring by a teacher of the center. The data registry form was specifically developed for the study and was the same in both phases of the study. A single dentist examined all the children.

An analysis was made of the evolution of the anteroposterior occlusal parameters in both the deciduous dentition and in the mixed and permanent dentition. Specifically, the following was evaluated in the anteroposterior plane:Canine occlusion: class I occlusion was considered when the cuspid of the upper canine occluded between the lower canine and first premolar, and in the absence of the canine, with the deciduous lower first molar; class II occlusion in turn was considered when occlusion was cuspid to cuspid, or when the distal aspects of the upper canines occluded anterior to the mesial aspects of the lower canines; and class III occlusion was considered when the upper canines occluded behind the lower canines.Molar occlusion: class I occlusion was considered when the mesiovestibular cuspid of the upper first molar occluded in the same plane as the central groove of the lower first molar; class II occlusion in turn was considered when the mesiovestibular cuspid of the upper first molar occluded anterior to the central groove of the lower first molar; and class III occlusion was considered when the mesiovestibular cuspid of the upper first molar occluded behind the central groove of the lower first molar.Terminal plane: this is the relation, in the anteroposterior plane, of the perpendicular planes formed from the distal surface of the deciduous upper second molar to the distal surface of the deciduous lower second molar, when positioned in occlusion.

There are three different terminal planes: (a) flush terminal plane, when the deciduous upper and lower second molars with their cuspids in opposition form a single posterior plane; (b) mesial terminal plane or mesial step, when the plane of the deciduous lower second molar lies anterior to the upper second molar, forming a step to the mesial side; and (c) distal terminal plane or distal step, when the plane of the deciduous lower second molar lies behind the upper second molar, forming a step to the distal side.

All these anteroposterior occlusal parameters were analyzed in the children by just one explorer, and were performed by direct clinical vision.

### Statistical Analysis

A general descriptive analysis of the children was made, with cross-comparisons of the qualitative variables. The chi-square test was applied to identify parameters with significant differences, using Haberman adjusted residues, to independently determine the significance of the differences in percentage values with respect to the total sample.

The data were entered in MS Excel spreadsheets, and the SPSS version 26 statistical package was used for analysis. Statistical significance was considered for *p* < 0.05.

## 3. Results

The collected data are descriptively presented in the form of tables and figures, and are expressed according to the evolution of the anteroposterior relation from deciduous dentition to mixed-permanent dentition. Specifically, a comparative description is made (deciduous dentition-mixed and permanent dentition) of the anteroposterior parameters, together with a description of the relations and evolutive changes of the terminal plane from deciduous dentition to mixed and permanent dentition, and the reporting of molar occlusion in mixed and permanent dentition (second phase).

### 3.1. Anteroposterior Plane

#### 3.1.1. Canine Occlusion

Upon analyzing canine class jointly and separately on both sides and in both the deciduous dentition (phase 1) and in the mixed-permanent dentition (phase 2), the deciduous dentition in most children was seen to be characterized by bilateral class I canine occlusion (75.5% of the cases), while 13.6% and 1.6% presented classes II and III occlusion, respectively.

Asymmetry or disparity in canine occlusion on both sides (right and left) was observed in 9.3% of the children in deciduous dentition (Figure 1). In relation to permanent canine occlusion (second phase of the study), the percentages were similar, except for a smaller percentage of children with canine class II occlusion (7.2%). The resulting asymmetries increased 14.3% and were more frequent in girls than in boys, though without reaching statistical significance (Figure 2). The distribution according to gender is shown in Table 1.

#### 3.1.2. Terminal Plane

The statistical analysis of the terminal plane on both sides and in the deciduous dentition (phase 1) and in the mixed-permanent dentition (phase 2) showed that in the deciduous dentition, 70% of the preschool children presented a bilateral flush terminal plane, 12.1% a mesial step, and 12.8% a distal step. The children with different terminal planes on one and the other side in turn accounted for 5.1% of the sample (Table 2).

In the second phase of the study, the abovementioned parameter was evaluated in mixed-permanent dentition, since at that time, most of the children still had the deciduous second molars in the mouth. In this phase, 27.2% presented a bilateral flush terminal plane, 6.7% a distal step, 45.1% a mesial step, and 21% had an asymmetrical terminal plane (Table 3).

Upon analyzing the evolution of the terminal plane from the deciduous to the mixed dentition, 23.1% of the children with a bilateral flush terminal plane in the deciduous dentition were seen to continue with a flush plane in the mixed dentition, while 6.6% evolved towards the mesial step, 5.2% towards the distal step, and 65.1% had evolved to a different plane (Figure 3).

By gender, and in both phase 1 and phase 2 of the study, we generally detected small and statistically nonsignificant differences between boys and girls (Table 4 and Table 5).

There were no significant differences upon analyzing the terminal planes in both phases in relation to socioeconomic status, with the exception of a flush step in the deciduous dentition, evaluated in the first phase of the study.

#### 3.1.3. Molar Occlusion in Permanent Dentition

Upon analyzing molar occlusion of the permanent first molars in the second phase of the study, the results showed that the most common molar occlusion corresponded to class I (64.5%), while 9.3% of the children presented half cusp molar class II, 2.8% full cusp class II (Figure 4), and the same percentage (2.8%) molar class III (Table 6).

Asymmetrical molar occlusion, with differences on both sides, was recorded in 20.6% of the children, and again was more frequent in girls than in boys (Table 7).

#### 3.1.4. Evolution of the Terminal Plane and Permanent Molar Occlusion

Consideration is required here of the relation between the different terminal planes (flush, mesial, and distal) of the deciduous second molars with the eruption and occlusion of the permanent first molars, since when the last teeth of the deciduous dentition (the second molars) erupt, the position of eruption of the permanent first molars is fully defined—and hence definitive correct or incorrect occlusion is established.

Upon examining this evolution between the terminal plane and occlusion of the permanent first molars, 55.3% of the children were seen to evolve from a flush terminal plane to correct anteroposterior molar class I occlusion of the permanent first molars; 11% evolved from a flush terminal plane to half cusp molar class II; and 2.4% evolved to full molar class II occlusion. On the other hand, upon examining the evolution of the mesial terminal plane, 2% of the children evolved to molar class I and 7.8% to molar class III occlusion. Lastly, in relation to the evolution of the distal terminal plane, 2% of the children were seen to present molar class I occlusion in the permanent dentition, while 2% presented half cusp class II and 0.4% full cusp class II occlusion. The rest (19.1%) presented asymmetrical terminal planes, i.e., the planes failed to coincide on the left and right side, due to different factors (Figure 5).

No statistically significant gender (Table 8) or socioeconomic differences were observed in the evolution of the terminal plane and anteroposterior occlusal relation of the permanent first molars.

## 4. Discussion

One of the main problems in longitudinal studies is the localization of patients that were previously studied. In effect, it is very difficult if not impossible to locate or secure the cooperation of many children, due to different reasons (change of address of the family, change in school, rejection to participate in a new study, the start of orthodontic treatment, etc.).

According to the literature, Hellman was the first to publish a study on the evolution of the dental arches, and Lewis in a longitudinal study reported that there is hardly no growth of the dental arches between 3–5 years of age—growth taking place from the age of 6 to 12 years [14,15].

Most authors coincide that the most important dimensional changes of the arches take place during the eruptive periods (mixed dentition). This is logical, taking into account that the teeth begin to need space as the dentition moves from deciduous to permanent. In this regard, Facal concluded that there are no significant changes in the dimensions of the arches during the deciduous dentition phase, and that their growth rate is independent of that of other structures [16].

With regard to the terminal plane and molar occlusion, at two and a half years of age, when the deciduous second molars erupt, an occlusal plane is formed that, according to Baume, may or may not be slightly tilted downwards and mesial [17]. In this sense, development of the arches remains stable until eruption of the lower incisors and permanent first molar at 6 years of age [16].

In our study, most of the children presented a flush terminal plane on both sides (70%), as a coincidence with the observations of Bhat et al. and Anu et al., and a minority (5.1%) showed asymmetrical terminal planes, due to different factors (agenesis, retention, or even tooth extraction) [13,18].

Hegde, in a study of children 3–4 years of age, also observed a predominance of flush step and canine class I occlusion, while the mesial step was seen to be predominate in children 4–5 years of age [19].

### Anteroposterior Changes of the Arches

According to Font, canine occlusion is more stable than the terminal plane and molar occlusion, since the canine erupts and remains in a stable condition without being influenced by other modifying factors [20]. Our results showed most children to have right and left canine class I occlusion (75.5%), while 13.6% presented class II and 1.6% class III occlusion. A disparity of canine occlusion on both sides was seen in 9.3% of the children in the deciduous dentition.

Sharma, in a study of children between 3–5 years of age, recorded a predominant percentage of canine class I occlusion and a flush terminal plane [21]. In turn, although Albakri analyzed older children, the majority likewise presented canine class I occlusion (68%), followed by classes II and III [22]. Coinciding with the abovementioned authors and our own data, Sotiria and Golovachova found a majority of children who presented canine class I occlusion with a flush terminal plane [23,24]. Additionally, Sivakumar found the higher prevalence of flush terminal plane in mixed dentition in females and males [25].

Knowing the evolution of occlusion in the anterior-posterior direction from the terminal plane and the canine occlusion of the deciduous dentition to the molar and canine relationship of the mixed and permanent dentition is important in order to guide, prevent, and intercept possible malocclusions in the permanent dentition.

The present study has limitations. In effect, a larger sample would be needed in order for the results to be more significant. However, over such a large time period between one phase and the next, many cases that were studied in the first phase are inevitably lost, due to the reasons mentioned above. On the other hand, the fact that very few longitudinal studies of the anteroposterior relations from deciduous dentition to mixed-permanent dentition have been published implies that our results cannot be contrasted with those of other investigators.

## 5. Conclusions

Based on the results obtained in the present study, it can be concluded that the most common bilateral terminal plane in deciduous dentition was a flush plane (70%), followed by distal and mesial steps, with few differences between them. The flush plane most often evolved towards a mesial plane during the mixed dentition transition period, and subsequently towards molar class I occlusion. The mesial terminal plane evolved towards molar class III in the permanent first molars and the distal terminal plane towards class II. Canine class I occlusion was the most frequent presentation in both the deciduous dentition and in the permanent dentition. Knowing the age range in which maximum dental development and growth in both arches occurs may contribute to prevent malocclusions and the possible need for orthodontic-orthopedic treatment, resulting in improved outcomes and greater stability.

A flush terminal plane is the most frequent presentation, and may determine class I occlusion or half cusp or full cusp class II occlusion of the permanent first molars, depending on space considerations and anterior mandibular displacement.

## Figures and Tables

**Figure 1 children-10-01708-f001:**
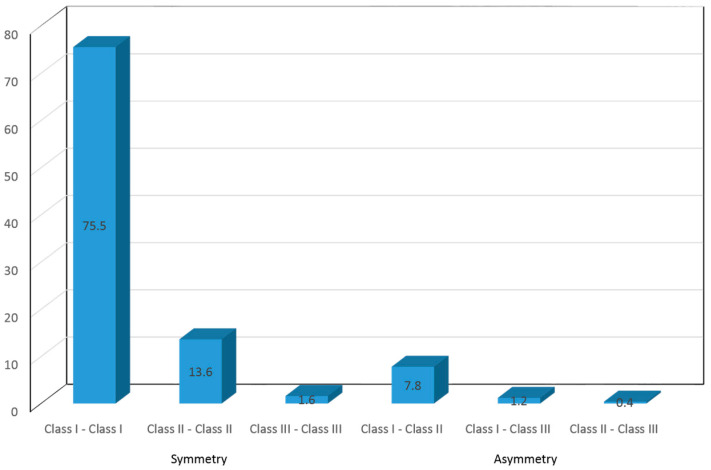
Canine occlusion in deciduous dentition (phase 1).

**Figure 2 children-10-01708-f002:**
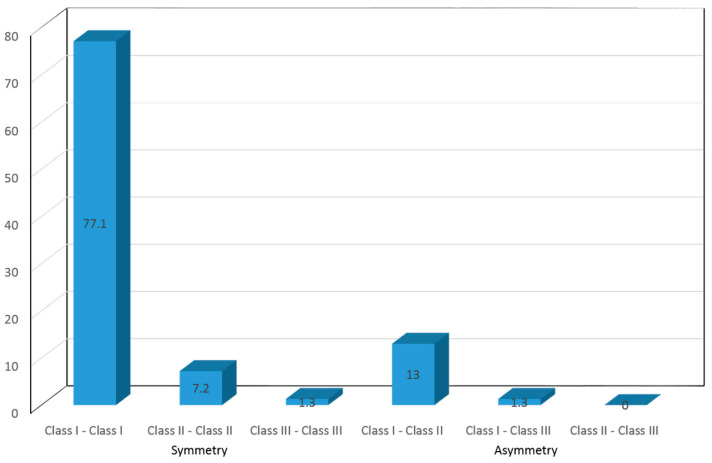
Canine occlusion in mixed—permanent dentition (phase 2).

**Figure 3 children-10-01708-f003:**
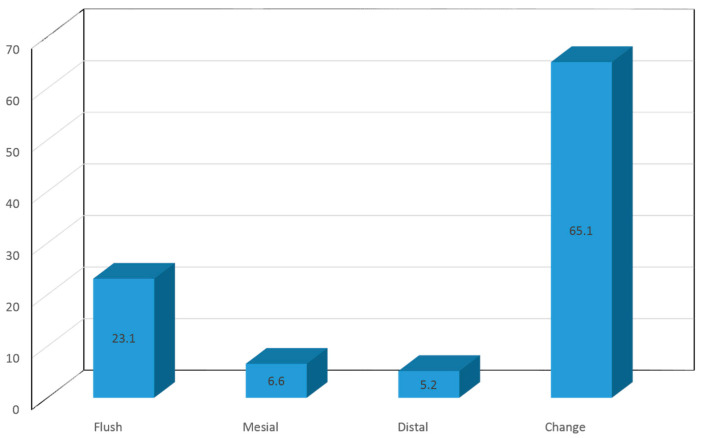
Evolution of the terminal plane over time (phase 1–2).

**Figure 4 children-10-01708-f004:**
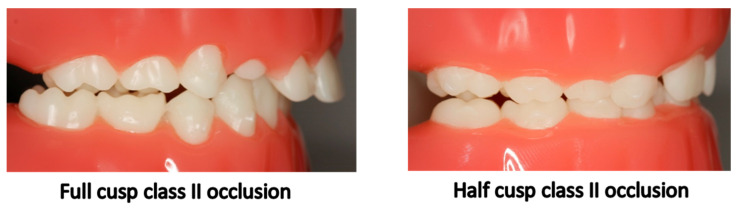
Check of the class II occlusion.

**Figure 5 children-10-01708-f005:**
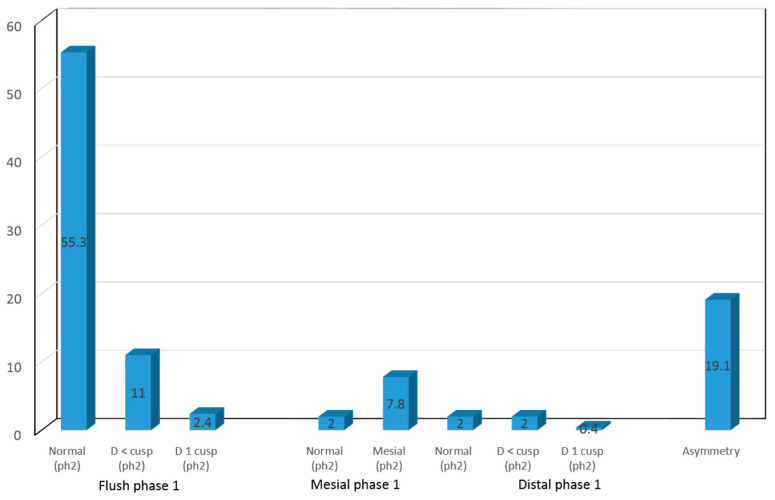
Terminal plane phase 1—Anteroposterior relations phase 2.

**Table 1 children-10-01708-t001:** Canine occlusion. Gender distribution.

Variable	Categories	Female		Male		Sig.
		Frequencies	Percentages	Frequencies	Percentages	
Canine occlusion (Phase 1)	Classes I–I	122	74.4	72	77.4	N.S.
	Classes II–II	24	14.6	11	11.8	
	Classes III–III	2	1.2	2	2.2	
	Classes I–II	14	8.5	6	6.5	
	Classes I–III	2	1.2	1	1.1	
	Classes II–III	0	0.0	1	1.1	
Canine occlusion (Phase 2)	Classes I–I	127	77.5	71	76.5	N.S.
	Classes II–II	11	6.3	8	8.7	
	Classes III–III	2	1.4	1	1.2	
	Classes I–II	23	14.1	10	11.1	
	Classes I–III	1	0.7	3	2.5	
	Classes II–III	0	0.0	0	0.0	

**Table 2 children-10-01708-t002:** Distribution of the terminal plane in the deciduous dentition (phase 1).

Variable	Categories	Frequencies	Percentages
Terminal plane (Phase 1)	Flush–Flush	180	70.0
	Mesial–Mesial	31	12.1
	Distal–Distal	33	12.8
	Flush–Mesial	1	0.4
	Flush–Distal	12	4.7
	Mesial–Distal	0	0.0

**Table 3 children-10-01708-t003:** Distribution of the terminal plane in the mixed-permanent dentition (phase 2).

Variable	Categories	Frequencies	Percentages
Terminal plane (Phase 2)	Flush–Flush	70	27.2
	Mesial–Mesial	116	45.1
	Distal–Distal	17	6.7
	Flush–Mesial	29	11.2
	Flush–Distal	15	5.8
	Mesial–Distal	10	4.0

**Table 4 children-10-01708-t004:** Distribution of the terminal plane by gender (phase 1).

Variable	Categories	Female		Male		Sig.
		Frequencies	Percentages	Frequencies	Percentages	
Terminal plane (Phase 1)	Flush–Flush	113	68.9	67	72.0	N.S.
	Mesial–Mesial	20	12.2	11	11.8	
	Distal–Distal	22	13.4	11	11.8	
	Flush–Mesial	0	0.0	1	1.1	
	Flush–Distal	9	5.5	3	3.2	
	Mesial–Distal	0	0.0	0	0.0	

**Table 5 children-10-01708-t005:** Distribution of the terminal plane by gender (phase 2).

Variable	Categories	Female		Male		Sig.
		Frequencies	Percentages	Frequencies	Percentages	
Terminal plane (Phase 2)	Flush–Flush	46	27.7	25	26.5	N.S.
	Mesial–Mesial	73	44.5	43	46.0	
	Distal–Distal	11	6.6	6	6.9	
	Flush–Mesial	19	11.7	10	10.3	
	Flush–Distal	7	4.4	7	8.0	
	Mesial–Distal	8	5.1	2	2.3	

**Table 6 children-10-01708-t006:** Permanent dentition molar class (phase 2).

Variable	Categories	Frequencies	Percentages
Anteroposterior relations (Phase 2)	CI	166	64.5
	CII < one cusp	24	9.3
	CII one cusp	7	2.8
	CIII	7	2.8
	Asymmetry	53	20.6

**Table 7 children-10-01708-t007:** Permanent dentition molar class (phase 2). Gender distribution.

Variable	Categories	Female		Male		Sig.
		Frequencies	Percentages	Frequencies	Percentages	
Anteroposterior relations (Phase 2)	CI	105	64.2	61	65.3	N.S.
	CII < one cusp	16	9.4	8	9.0	
	CII one cusp	5	3.1	2	2.2	
	CIII	5	3.1	2	2.2	
	Asymmetry	33	20.2	20	21.3	

**Table 8 children-10-01708-t008:** Terminal plane (phase 1)—anteroposterior relations (phase 2). Gender distribution.

Variable	Categories	Female		Male		Sig.
		Frequencies	Percentages	Frequencies	Percentages	
Terminal plane (Phase 1)—Anteroposterior relations (Phase 2)	Flush–CI	81	49.1	48	51.7	N.S.
	Mesial–CI	14	8.2	6	6.7	
	Distal–CII	9	5.7	3	3.4	
	Asymmetry–CII	2	1.3	3	3.4	
	Flush–Other	33	20.1	18	19.1	
	Mesial–Other	5	3.1	5	5.6	
	Distal–Other	13	8.2	9	9.0	
	Asymmetry–Other	7	4.4	1	1.1	

## Data Availability

The entire dataset on which the conclusions of the study are based is presented in the principal manuscript. The related information is available upon request. You can request it by email: adominre@us.es. The data are not publicly available due to privacy and ethical restrictions.

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
