# Peer review of "Evolution of the Terminal Plane from Deciduous to Mixed Dentition"

_children, 2023, doi:10.3390/children10101708_

Round 1

Reviewer 1 Report

The aim of the study was to evaluate how the terminal plane and canine relationship in primary dentition transitioned into mixed/ permanent dentition. The authors did a good job. But the methods section and presentation of results could be improved. Especially, the tables were hard to follow. Many details were missing in the methods section. Please refer to the below comments and make necessary changes.

Specific comments:

Methods:

·         Line 66: From how many total schools were these 18 schools selected? Please provide few more details about the randomization technique followed

·         Line 67-68: How were socio-economic factors considered: low, medium and high? What was the basis of the classification?

·         Line 99-101: Would recommend adding more information on how was the terminal plane examined exactly? Is it direct vision or indirect vision?

·         Line 103: Is it “in the absence of canine” or “in the absence of first premolar”. Please verify

·         Line 107-111: Please explain what is the difference between half cusp molar and full cusp molar class II malocclusion. Please provide any schematic diagrams or clinical pictures.

·         Line 126: So, was ANOVA (parametric test) used to analyze any data?

·         Line 125-129: Looks little confusing. What tests were used exactly?

Results:

·         Line 178-179: “65.1% had evolved to a different plane (Figure 3).” What does different plane include?

·         Line 186 & Line 213: What socio-economic factors were considered in the analysis? What statistical tests were used to analyze socio-economic factors. Please present the results from the analysis.

Discussion:

·         Line 246: “most children to have right and left canine class II occlusion (75.5%)” Isn’t it class I occlusion? Data presented in the study contradicts this statement.

Figure 1 & 2:

o   Title needs to be updated to make it clear what is being measured exactly? So, is it describing the canine relation on the right and left side in the deciduous dentition in figure 1 and mixed-permanent dentition in figure 2?

o   Axis titles seem to be not in English

o   Update the graphs with the actual numbers as well.

·         Axis labels in all the figures are not in English

Table 1, 4, 5, 7 & 8:

o   Would recommend moving the percentages column next to the counts to make it easy to follow.

o   Would recommend adding the actual p-values instead of just saying “NS”

The aim of the study was to evaluate how the terminal plane and canine relationship in primary dentition transitioned into mixed/ permanent dentition. The authors did a good job. But the methods section and presentation of results could be improved. Especially, the tables were hard to follow. Many details were missing in the methods section. Please refer to the below comments and make necessary changes.

Specific comments:

Methods:

·         Line 66: From how many total schools were these 18 schools selected? Please provide few more details about the randomization technique followed

·         Line 67-68: How were socio-economic factors considered: low, medium and high? What was the basis of the classification?

·         Line 99-101: Would recommend adding more information on how was the terminal plane examined exactly? Is it direct vision or indirect vision?

·         Line 103: Is it “in the absence of canine” or “in the absence of first premolar”. Please verify

·         Line 107-111: Please explain what is the difference between half cusp molar and full cusp molar class II malocclusion. Please provide any schematic diagrams or clinical pictures.

·         Line 126: So, was ANOVA (parametric test) used to analyze any data?

·         Line 125-129: Looks little confusing. What tests were used exactly?

Results:

·         Line 178-179: “65.1% had evolved to a different plane (Figure 3).” What does different plane include?

·         Line 186 & Line 213: What socio-economic factors were considered in the analysis? What statistical tests were used to analyze socio-economic factors. Please present the results from the analysis.

Discussion:

·         Line 246: “most children to have right and left canine class II occlusion (75.5%)” Isn’t it class I occlusion? Data presented in the study contradicts this statement.

Figure 1 & 2:

o   Title needs to be updated to make it clear what is being measured exactly? So, is it describing the canine relation on the right and left side in the deciduous dentition in figure 1 and mixed-permanent dentition in figure 2?

o   Axis titles seem to be not in English

o   Update the graphs with the actual numbers as well.

·         Axis labels in all the figures are not in English

Table 1, 4, 5, 7 & 8:

o   Would recommend moving the percentages column next to the counts to make it easy to follow.

o   Would recommend adding the actual p-values instead of just saying “NS”

Author Response

Dear reviewer:
Thank you very much for your comments. As you have suggested, we have made the changes you have indicated in the manuscript.
Best regards.

Reviewer 2 Report

The article is not on the template of the journal.

The present study evaluates the evolution of the 16 terminal plane and canine occlusion class in the same children from deciduous to mixed dentition. 

Figure one is inserted in the text and does not have a title.

Provide bibliography for this paragraph- One of the main problems in longitudinal studies is the localization of patients that were previously studied. In 221 effect, it is very difficult if not impossible to locate or secure the cooperation of many children, due to different 222 reasons (change of address of the family, change in school, rejection to participate in a new study, the start of 223 orthodontic treatment, etc.)

The  results and discussion chapter has to be more stuctured and in the discussion please include more recent published articles.

Moderate

Author Response

We thank you for all your comments. We have endeavoured to respond to all of them. We hope you will be satisfied with our answers. Thank you very much. Best regards.

Reviewer 3 Report

Dear Authors, this paper about "evolution of the terminal plane from deciduous to mixed teeth" is really interesting and well written. I am pretty sure that both researchers and dental professionals will find it very useful for their purposes.

Some issues need to be solved before its final publication in the Journal.

Title: please avoid capital letters

Abstract: Please divide abstract into "introduction", "materials and methods", "results", "conclusion".

Introduction: This is a really important part of an article, it helps the reader to deep into the subject of the paper. In your paper this part is really short; you should add some more information especially regarding the presence of systemic factors that might influence dental change such as caries; this paper could help: -        Efficacy of Two Toothpaste in Preventing Tooth Erosive Lesions Associated with Gastroesophageal Reflux Disease. Ludovichetti, F.S.; Zambon, G.; Cimolai, M.; Gallo, M.; Signoriello, A.G.; Pezzato, L.; Bertolini, R.; Mazzoleni, S. 

Appl. Sci. 2022, 12, 1023. 

Materials and methods and results section: this part is well written

Discussion: The discussion should delve deeper into the significance of the study's findings. It should explain why the results are important and how they contribute to the existing body of knowledge in the field. Additionally, the limitations mentioned at the end should be discussed earlier in the discussion section to provide context for the findings.

Author Response

(The authors gave the same response as above.)

Round 2

Reviewer 1 Report

Few of the comments even though were responded through author responses, no changes were made in the actual manuscript. So, would recommend to make necessary changes based on the following comments.

·       Line 74-76: Please update more information about the randomization technique followed in the study. The response provided could be added into the manuscript

·       Please add information about how the socio-economic levels were considered -low, medium and high. The author response regarding it was not clear enough- “level of education of the parents, and the type of school (private, charter or public)”. So, what was considered high socio-economic status vs low socio-economic status exactly?

·       Please update the manuscript with how did the terminal plane was examined. “Direct clinical vision”

·       Would still recommend adding some schematic diagrams or clinical pictures to showing different canine and molar relationships. Especially- half cusp or full cusp class II occlusion

·       If most of the data showed non-normal distribution, why was ANOVA used to measure means values between the groups. What means values are being analyzed exactly?

·       Line 136-140- Repetition of texts.

Author Response

Thank you for your comments. We are sure that they contribute to improve the quality of the article.
